# Amphiphilic Poly(vinyl Alcohol) Copolymers Designed for Optical Sensor Applications—Synthesis and Properties

**Katerina Lazarova** [1,*], **Silvia Bozhilova** [2], **Christo Novakov** [2], **Darinka Christova** [2] and **Tsvetanka Babeva** [1,*]

1   Institute of Optical Materials and Technologies "Akad. J. Malinowski", Bulgarian Academy of Sciences, Akad. G. Bonchev str., bl. 109, 1113 Sofia, Bulgaria

2   Institute of Polymers, Bulgarian Academy of Sciences, Akad. G. Bonchev Str., bl. 103-A, 1113 Sofia, Bulgaria; s.bozhilova@polymer.bas.bg (S.B.); hnovakov@polymer.bas.bg (C.N.); dchristo@polymer.bas.bg (D.C.)

*   Correspondence: klazarova@iomt.bas.bg (K.L.); babeva@iomt.bas.bg (T.B.); Tel.: +359-2-979-3526 (K.L.)

**Abstract:** A possible approach for enhancement of Poly(vinyl alcohol) (PVA) humidity-sensing performance using hydrophobically modified PVA copolymers is studied. Series of poly(vinylalcohol-*co*-vinylacetal)s (PVA–Ac) of acetal content in the range 18%–28% are synthesized by partial acetalization of hydroxyl groups of PVA with acetaldehyde and thin films are deposited by spin-coating using silicon substrates and glass substrates covered with Au–Pd thin film with thickness of 30 nm. Sensing properties are probed through reflectance measurements at relative humidity (RH) in the range 5%–95% RH. The influence of film thickness, post-deposition annealing temperature, and substrate type/configuration on hysteresis, sensitivity, and accuracy/resolution of humidity sensing is studied for partially acetalized PVA copolymer films, and comparison with neat PVA is made. Enhancement of sensing behavior through preparation of polymer–silica hybrids is demonstrated. The possibility of color sensing is discussed.

**Keywords:** poly(vinyl alcohol) copolymers; thin films; humidity sensing; optical sensors

## 1. Introduction

Poly(vinyl alcohol) (PVA) is a hydrophilic and very water-soluble polymer due to pendant hydroxyl groups, which are mainly responsible for its reactivity and crystallinity [1]. Due to its outstanding mechanical and film-forming properties PVA is used in a variety of areas such as membranes, adhesives, coatings, etc. [2]. Because PVA can absorb and desorb water quickly, an increasing interest in the research of PVA-based humidity sensors has been observed recently and PVA is implemented as a humidity-sensitive medium in various sensor types [3–8].

However, development of PVA-based optical humidity sensors with high sensitivity, wide dynamic range and linearity, stability, and low hysteresis is still a challenge. This is probably due to the highly water-soluble nature of PVA that limits its stable sensing properties in a form of nanometer-thick polymer film. One possible approach to overcome these drawbacks is to use different composites consisting of PVA as a matrix. Mixture of PVA and graphene quantum dots (GQDs) [9–11] and crosslinked PVA/functionalized graphene oxide nanocomposite films [12] were used for humidity-sensing using optical fiber technology [9–11] and attenuated reflectance measurements [12]. Composites consisting of PVA and nanosilica particles were used for humidity sensing through crystal microbalance [13] or for depositing sensitive opal structures [14]. Polyaniline/poly(vinyl alcohol) composites [15] and silver–polyaniline/polyvinyl alcohol composites [16] were used as sensitive media for acoustic

wave-impedance humidity sensors and resistive humidity sensors, respectively. Thick PVA substrates (around 80 microns) doped with silver nanoparticles were applied for humidity sensing through transmittance measurements [17].

Another possible approach to enhance PVA sensing performance is to use hydrophobically modified PVA copolymers such as poly(vinyl acetal)s [18]. In general, poly(vinyl acetal)s are class of polymers obtained by reaction of PVA with aldehydes, especially formaldehyde, acetaldehyde, and butyraldehyde finding advanced application as structural adhesives in the aircraft industry, as the interlayer in automotive safety glass, etc. [19]. The contents of unreacted hydroxyl groups along with the acetal rings and the molecular weight determine the polymer properties. Hydrophilicity of partially acetalized PVA is reduced while maintaining its inherent response to humidity.

Our previous studies [20,21] have shown that hydrophobically modified PVA copolymers, namely poly(vinylalcohol-*co*-vinylacetal)s (PVA–Ac) of acetal content in the range 18%–28% prepared as a single films on opaque substrate are suitable for optical sensing of humidity. In this work, the influence of film thickness, post-deposition annealing and substrate type/configuration on humidity-sensing properties is studied for partially acetalized PVA films and comparison with neat PVA is made. The possibility to improve the reaction toward humidity of the polymer thin films via doping with $SiO_2$ particles is explored and polymer–silica hybrids are obtained. Enhancement of sensing behavior through this approach is demonstrated and discussed.

## 2. Materials and Methods

### 2.1. Synthesis of PVA Copolymers

Series of hydrophobically modified PVA copolymers were synthesized by partial acetalization of hydroxyl groups of PVA (average polymerization degree 1600) with acetaldehyde following the procedure described elsewhere [18]. The copolymer composition of obtained PVA–Ac, namely the content of acetal groups, was estimated by using Nuclear Magnetic Resonance (NMR) spectroscopy. [1]H NMR spectra were taken on a Bruker Avance DRX 250 spectrometer (Bruker Corporation, Billerica, MA, USA) in DMSO-$d_6$ as solvent. The extent of hydrophobic modification was evaluated by means of UV-VIS spectroscopy. Transmittance of copolymer aqueous solutions was studied at wavelength of 500 nm at concentration of 5 g/L as a function of temperature. Cloud points ($T_{CP}$) of copolymers solutions were determined from the measured transmittance-vs-temperature curves as the temperature at transmittance level of 50%.

Copolymer solutions of 1 wt.% concentration in mixed water–methanol solvent (20:80 volume ratio) were prepared for thin film deposition process. To obtain hybrid polymer–silica thin films, $SiO_2$ particles were in situ generated in copolymer solutions via the sol-gel method [22,23]. Calculated amount of the precursor tetraethyl orthosilicate (TEOS) was added to the copolymer solution under stirring and mixture was acidified with 1 M HCl to pH 2. The reaction mixture was homogenized by ultra-sonication for 30 min and then TEOS hydrolysis was continued at vigorous stirring on a magnetic stirrer at room temperature for 24 h. The obtained copolymer solution doped with $SiO_2$ particles was used for thin films deposition without further treatment. The condensation of the silica was completed during the annealing of the deposited thin hybrid films.

### 2.2. Deposition of Thin Films

Water–methanol solutions in a volume ratio of 20:80 and concentrations of 1 and 2 wt.% were used for deposition of acetal modified PVA films. Thin polymer films were deposited by spin-coating method at a rotation speed of 4000 rpm and time of 60 s using 0.250 mL of the solution. After deposition, the films were annealed in air for 30 min at 60 and 180 °C. For comparison, films of neat PVA were also prepared by depositing 2 and 5 wt.% water solution of PVA to achieve the same film thicknesses as in the case of modified films. Silicon wafers and Au–Pd covered optical glass plates were used as substrates. The Au–Pd sublayers with Au:Pd ratio of 80:20 and thickness of 30 nm were deposited on

glass substrates by cathode sputtering of gold/palladium target (Quorum Technologies, Lewes, UK) for 60 s under vacuum $4 \times 10^{-2}$ mbar using Mini Sputter Coater SC7620 system (Quorum Technologies, Lewes, UK).

Polymer thin films composites (polymer doped with $SiO_2$ particles) were spin-coated on silicon wafers (0.250 mL solution at concentration of 1 wt.%, 4000 rpm, 60 s) and post-annealed at 60 °C for 30 min.

*2.3. Characterization of Thin Films*

Optical constants (refractive index *n* and extinction coefficient *k*) and thickness of the films *d* were calculated simultaneously using previously developed two-stages nonlinear curve fitting method using measured reflectance spectra with UV-VIS-NIR (ultraviolet-visible-near infrared) spectrophotometer (Cary 5E, Varian, Australia) [24]. The sensing properties of the films were studied through recording reflectance spectra at different values of relative humidity (RH) in the range from 5% to 95% RH. The sample was placed in a quartz cell inside the spectrophotometer and the humidity decreased from ambient to 5% *RH* by purging dry argon in the cell. Then the recording of reflectance (or transmittance) value as a function of humidity was started. The continuous increase of humidity from 5% to 95% RH was achieved by bubbling argon through distilled water kept at 60 °C. In these experiments the reflectance/transmittance was measured at fixed wavelength that is preliminary chosen as the wavelength of the highest humidity responses. To determine this wavelength for each thin film ($\lambda_{max}$), along with optical constants and thickness (and its change), the reflectance spectra (320–800 nm) of the samples were measured at humidity of 5% and 95% RH in another set of humidity experiments and optical constants and thickness were determined.

To quantify and compare studied samples, three parameters were used. The sensitivity of the sensors, *S*, was calculated according to the following equation:

$$S = \frac{\Delta R}{RH_2 - RH_1}, \tag{1}$$

where $\Delta R$ (or $\Delta T$, if transmittance *T* is measured) is the change of film's reflectance (or transmittance) in % for humidity variation from $RH_1$ to $RH_2$. Accuracy/resolution ($\Delta RH$) of detection depends on the sensitivity and measurement accuracy in the signal and was calculated from:

$$\Delta RH = \frac{errR \; (\%)}{S \; (\%)}, \tag{2}$$

where *errR* = 0.3% (or *errT* = 0.1%, if *T* is measured) is the experimental error (accuracy) of *R* or *T* and *S* is the sensitivity, calculated by Equation (1).

Sometimes it is possible unwanted hysteresis to occur that is expressed in different values of *R* (or *T*) measured at the same values of humidity depending of the history of humidity, i.e., depending whether humidity increases or decreases. The percentage of hysteresis, *H* was determined through:

$$H(\%) = \frac{max \left| R_{up} - R_{down} \right|}{\Delta R_{max}} \cdot \frac{\Delta RH_{hyst}}{\Delta RH} \cdot 100, \tag{3}$$

where $R_{up}$ and $R_{down}$ are reflectance (or transmittance) values measured for increasing and decreasing humidity, respectively, $\Delta R_{max}$ is the reflectance (or transmittance) change in the whole range $\Delta RH$ of measured humidity and $\Delta RH_{hyst}$ is the humidity range where hysteresis is observed.

It is obvious from Equations (1)–(3) that the goal is to obtain the highest sensitivity and accuracy of detection and the lowest percentage of hysteresis.

## 3. Results

### 3.1. Characterization of Synthesized Polymers

Four PVA–Ac copolymers of different composition were synthesized varying PVA-to-acetaldehyde molar ratio. The reaction scheme and chemical structure of the obtained PVA copolymers are illustrated in Figure 1a. The copolymer composition and aqueous solution properties were studied by NMR and UV-VIS spectroscopy, respectively. The results are summarized in Table 1.

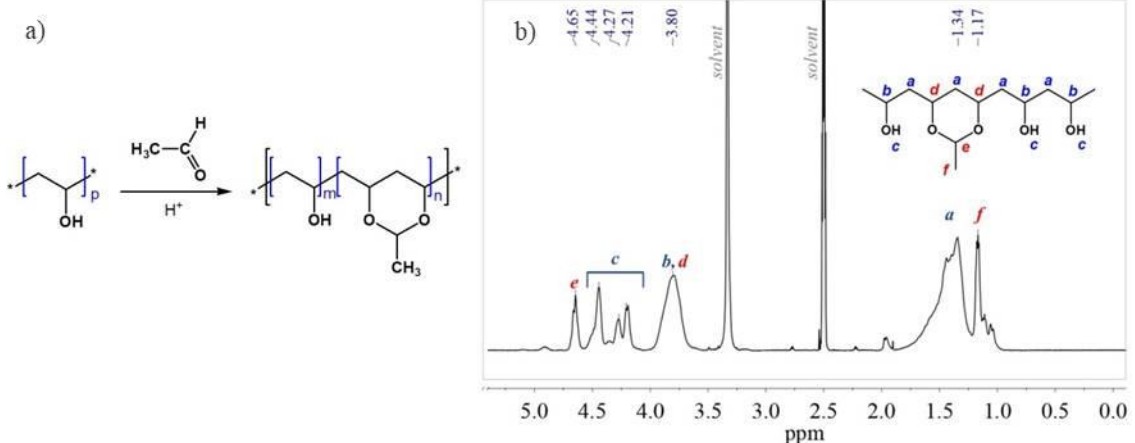

**Figure 1.** (**a**) Schematic presentation of acetalization reaction of PVA; (**b**) $^1$H NMR spectrum of copolymer AC18 (solvent DMSO-d$_6$).

**Table 1.** Acetal content and cloud point ($T_{CP}$) of modified PVA copolymers.

| Sample | Acetal Content, % (NMR) | $T_{CP}$ *, °C (UV-VIS) |
|---|---|---|
| PVA | 0 | -** |
| Ac18 | 18 | 47 |
| Ac19 | 19 | 40 |
| Ac24 | 24 | 30 |
| Ac28 | 28 | 27 |

\* As measured for 5 g/L aqueous copolymer solution; \*\* No $T_{CP}$ detected up to 90 °C.

Typical proton NMR spectrum of PVA–Ac is shown in Figure 1b. Copolymer composition expressed as a content of acetal groups was calculated by comparing the area of the peak assigned to the methine protons from the acetal group (e) to those assigned to the methine protons from the PVA main chain (b+d).

The synthesized PVA–Ac copolymers, although very water soluble at room temperature, undergo phase transition when increasing the temperature of the aqueous solutions and turn water insoluble. This is due to the introduced fractions of hydrophobic acetal groups and reflects in the reduced hydrogen bonding between copolymers and water molecules as compared to pure PVA. To evaluate the hydrophilic–hydrophobic balance of the copolymers, expected to influence the humidity-sensing properties of the corresponding thin films, $T_{CP}$ in dilute aqueous solutions were measured. Clouding curves of the copolymer aqueous solutions were registered, and $T_{CP}$ were estimated at 50% transmittance. As seen in Table 1, the higher the acetal content, the lower the $T_{CP}$.

### 3.2. Optimization of Thickness and Post-Deposition Annealing

When polymer films are exposed to humidity they change their thicknesses and refractive indices. This results in change of the measured reflectance or transmittance spectra. We have recently shown

that the dimensional change in polymer films with nanometer thickness in the range 100–400 nm depends on the initial thickness and increases with increasing thickness [25]. Figure 2a presents the change in thickness for studied samples (80 and 200 nm) at their exposure from low to high humidity. As expected, the degree of swelling of thicker films (200 nm) is substantially higher as compared to thinner films especially for 19% modified PVA films where the relative increase of thickness ($\Delta d/d$) is 120%. Furthermore, the post-deposition thermal treatment of samples at higher temperature (180 °C) does not lead to an improvement of swelling, as we have already shown in [21]. On the contrary, the dimensional changes of films pre-annealed at 180 °C are smaller as compared to those treated at 60 °C, especially for the neat PVA films which degree of swelling is almost 7 times lowered.

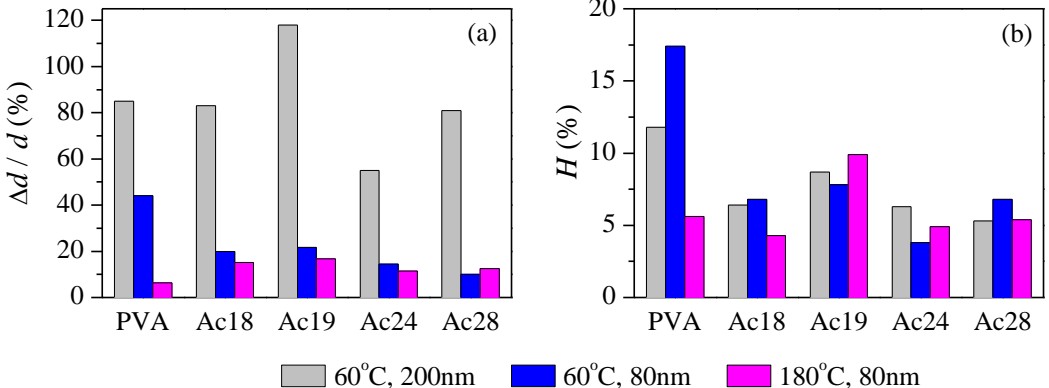

**Figure 2.** (**a**) Thickness change upon humidity exposure ranging from 5% to 95% RH of polymer films with thickness of 80 and 200 nm and of different acetal content pre-annealed at 60 and 180 °C; (**b**) Percentage of hysteresis, *H* calculated with eq. 3 for polymer films (80 and 200 nm) and of different acetal content pre-annealed at 60 and 180 °C.

It is well known that the hysteresis, *H*, is another very important parameter that determines the suitability of the material for sensor applications. The existence of *H* means measuring of different signal (reflectance or transmittance in our case) for the same humidity values depending whether humidity increases or decreases. It is obvious that *H* is unwanted parameter and the main goal is to keep its value as low as possible.

The hysteresis values *H* of all samples studied is summarized in Figure 2b and Table 2. A substantial decrease of hysteresis due to annealing at 180 °C is observed for neat PVA films. The smallest *H*-values (4.3% and 3.8%) are achieved for PVA-modified samples (80 nm) with acetal content of 18% and 24%, pre-annealed at 180 and 60 °C, respectively.

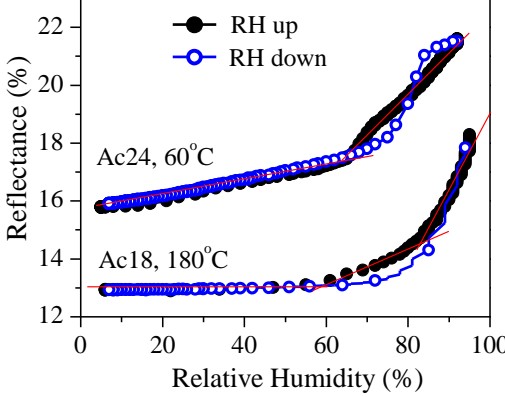

**Figure 3.** Reflectance versus relative humidity curves for films with 24% and 18% acetal content (80 nm thick), pre-annealed at 60 and 180 °C, respectively, measured for increasing (solid black symbols) and decreasing (open blue symbols) humidity.

**Table 2.** Post-annealing temperature ($T_{post}$), wavelength at which $R$ ($T$) measurements were conducted ($\lambda_{max}$), percentage of hysteresis ($H$), dynamic range, sensitivity, and accuracy for studied samples.

| Sample | $T_{post}$(°C) | $\lambda_{max}$ (nm) | $H$ (%) | Dynamic Range (% RH) | Sensitivity (%/% RH) | Accuracy (% RH) |
|---|---|---|---|---|---|---|
| PVA | 60 | 597 | 17.4 | <30 | <0.01 | >30 |
| PVA | 180 | 402 | 5.6 | <75 | 0.013 | 23 |
| Ac18 | 60 | 592 | 6.8 | <45 | 0.010 | 30 |
| Ac18 | 180 | 400 | 4.3 | >60 | 0.07 (60%–84% RH) 0.3 (>84% RH) | 4 1 |
| Ac19 | 60 | 400 | 7.8 | <40 | <0.01 | >30 |
| Ac19 | 180 | 400 | 9.9 | <60 | <0.01 | >30 |
| Ac24 | 60 | 408 | 3.8 | full | 0.03 (5%–65% RH) 0.14 (> 65% RH) | 10 2 |
| Ac24 | 180 | 406 | 4.9 | full | 0.01 (5%–60% RH) 0.02 (60%–77% RH) 0.08 (>77% RH) | 30 15 4 |
| Ac28 | 60 | 598 | 6.8 | <60 | < 0.01 | >30 |
| Ac28 | 180 | 600 | 5.4 | full | 0.01 (5%–70% RH) 0.05(>70% RH) | 30 6 |
| Ac24 ($T$%) | 60 | 460 | 3.6 | full | 0.03 (5%–70% RH) 0.14(>70% RH) | 3 0.7 |
| Ac24p1 | 60 | 424 | 1.7 | full | 0.03 | 10 |
| Ac24p2 | 60 | 482 | 3.9 | full | 0.02 | 15 |

　　　One can conclude that the most suitable samples are Ac24 and Ac18 annealed at 60 and 180 °C, respectively. The dependence of reflectance on relative humidity in the range 5%–95% RH ($R$-vs-RH curves) for both samples of the smallest H-values are presented in Figure 3.

　　　It is seen that reflectance for Ac18 is almost the same in wide humidity range (5%–60% RH) and starts to increase exponentially at RH > 60%. On the contrary, for sample Ac24 two linear dependences of $R$-vs-RH plot with different slopes are well distinguished. The sensitivity is 0.03 at RH = 5%–65% and increases to 0.14 at RH > 65%. Considering that the measurement error in reflectance is 0.3% the accuracy of humidity measurement is 10% and 2% RH, respectively (Table 2). We should note that although the swelling is the strongest for thick films (200 nm) they are not suitable for sensing because exhibit high values of hysteresis (Figure 2b). Additional measurements of reflectance as a function of relative humidity (not shown) demonstrated that thicker samples have narrow dynamic range. Generally, for thicker samples $R$ changes only for humidity higher than 80% RH. For sample Ac19 the case is even worse because there is unambiguity—one and the same reflectance values are measured for different humidity. The reason is the periodicity of the dependence of R on d. When the change of thickness due to humidity is higher than the period of $R$-vs-$d$ dependence then a periodicity in $R$-vs-RH curve could be observed. Usually this unambiguous behavior appears for thicker films ($d$ higher than 150 nm) where swelling is stronger as compared to thinner ones [25].

　　　Considering all results presented above we concluded that the most appropriate sample for our purposes is Ac24 (24% acetal content PVA) with approximate thickness of 80 nm pre-annealed at 60 °C. Further efforts are concentrated on optimization of sensing properties of this material using two approaches: (i) humidity sensing through transmittance measurements; and (ii) doping with $SiO_2$ particles.

### 3.3. Humidity Sensing Using Transmittance Measurements

It is well known that in general case measuring the transmittance is easier, more accurate, and less expensive than measuring the reflectance. Therefore, it will be more advantageous to use transmittance measurements as optical read-out for detecting humidity. To perform transmittance measurements, the sensitive medium should be deposited on transparent substrate. Usually these are glass or plastic with approximate refractive index in the range 1.4–1.5 that is very close to refractive index of the polymers used for detection. Thus, the small optical contrast will lead to low sensitivity of detection, because it will be difficult to distinguish the thin film from the substrate because of the match of their refractive indices. We have already shown that quarter-wavelength multilayers stacks (Bragg stacks) and glass covered with thin semitransparent metal overlayer are suitable transparent substrates for optical detecting of humidity in transmittance mode [25,26]. When planar Bragg stacks are used for substrates, the sensitivity of detection increases with thickness of the sensitive medium deposited on top and it is the highest for films thicker than 250 nm [25]. However, as already mentioned an ambiguity exists when films with thicknesses higher than 100–150 nm are used as sensitive media. Therefore, in this study we use thin film with thickness of 80 nm deposited on Au–Pd covered glass substrate. The thickness of the metal overlayer is selected to be 30 nm thus guaranteeing transmittance to be around 50%.

The transmittance of Ac24 thin film with thickness of 80 nm, deposited on glass covered with Au–Pd overlayer with thickness of 30 nm as a function of relative humidity is shown in Figure 4. The observed percentage of hysteresis of 3.6% is very close to the value obtained when reflectance as a function of humidity is used (3.8%) (Figure 2b). Furthermore, similarly to the case of reflectance measurements (Figure 2b), two well-distinguished linear parts of the *T*-vs-RH curve are observed. The calculated sensitivities are comparable to the case of silicon substrate: 0.03 in the range 5%–70% *RH* and 0.14 for *RH* > 70%. However, because of the higher accuracy in transmittance measurements (errT is 0.1% as compared to errR = 0.3% when R is measured) the accuracy of sensing ΔRH is 3 times higher (Equation (2)). Thus, using the configuration polymer film/metal layer/glass, less than 1% RH could be distinguished in the range of high humidity (*RH* > 70%) and 3% for RH < 70% (Table 2).

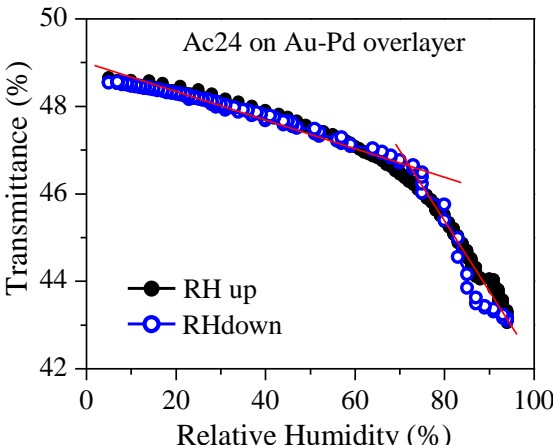

**Figure 4.** Transmittance versus relative humidity curve for PVA film (80 nm thick) of 24% acetal content, deposited on Au–Pd (30nm) covered glass substrate, pre-annealed at 60 °C measured for increasing (solid black symbols) and decreasing (open blue symbols) humidity.

### 3.4. Doping with SiO₂ Particles

Incorporation of silica nanoparticles may further enhance the sensing properties of studied PVA–Ac copolymer thin films. The silanol groups on the particles' surface can develop intermolecular bonds with PVA hydroxyl groups and may assist improving the sensing performance and reducing

hysteresis. This approach was demonstrated by implementing in situ sol-gel reaction of TEOS in Ac24 copolymer solutions prior to the thin film deposition.

The influence of $SiO_2$ particles doping on humidity thickness change and percentage of hysteresis is illustrated in Figure 5a. The comparison between undoped (Ac24) and differently doped samples (Ac24p1, 20% and Ac24p2, 50%) shows that the swelling ($\Delta d/d$) due to humidity exposure decreases with doping from 14.5% for undoped sample to 6.6% and 4.2% for 20% and 50% doped samples, respectively. The possible reason is the rigidity of the films that increases when $SiO_2$ particles are incorporated in the polymer matrix. However, the doping has a positive effect on percentage of hysteresis: H decreases more than twice for 20% doped sample (Ac24p1): from 3.8% (undoped film) to 1.7% (20% doped film). In this case, the increased rigidity of the doped film contributes positively because it prevents fast shrinking of polymer films during humidity desorption in high range (RH > 70%) where the hysteresis is commonly observed.

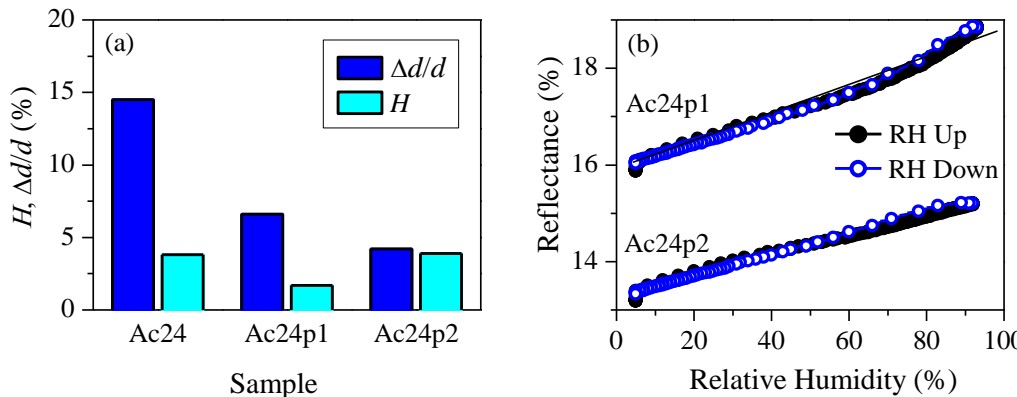

**Figure 5.** (**a**) Percentage of hysteresis, *H* (Equation (3)) and relative change of film thickness $\Delta d/d$ due to humidity exposure from 5% to 95% RH for polymer films (about 80 nm in thickness) with 24% acetal content (Ac24) doped with 20% (Ac24p1) and 50% (Ac24p2) $SiO_2$ particles; (**b**) Reflectance versus relative humidity curve for films with 24% acetal content doped with 20% and 50% $SiO_2$ particles measured for increasing (solid black symbols) and decreasing (open blue symbols) humidity.

The *R*-vs-RH curves for the doped films are plotted in Figure 5b. A very good linear dependence of measured reflectivity is observed in the whole RH range for both samples, mostly pronounced for heavily doped one (Ac24p2) sample. This means that the acetal modified PVA films doped with appropriate amount of $SiO_2$ particles are very suitable for optical sensing of humidity offering linearity in the entire humidity range.

Results summary is presented in Table 2. From all studied samples the smallest hysteresis is observed for Ac24 polymer film doped with 20% $SiO_2$ particles. The sample has full dynamic range and linear dependence of the measured signal as a function of relative humidity. The accuracy of sensing is 10% RH but could be substantially increased if transmittance measurements are implemented. More studies are underway to optimize the sensing behavior of doped polymer samples.

It is seen from Table 2 that another good option for optical sensing of humidity is Ac24 polymer film (PVA with 24% acetal content) deposited on glass substrate with Au–Pd overlayer (30 nm). At RH > 70% the accuracy is less than 1% RH, the dynamic range is wide, and the hysteresis is acceptable.

## 3.5. Color Sensing of Humidity

When transparent film is deposited on absorbing substrate (silicon wafer or metal overlayer in our case) it exhibits a certain color which depends on film's optical thickness. When optical thickness changes due to adsorption of water vapors (as is in our case) from the environment, the reflectance spectrum of the film changes and the color alters. Optical sensing based on perceptual color change in response to analyte of interest offers simplicity and is preferred in color sensing of vapors. So the next

step of our investigation was to study the possibility of color sensing of humidity, i.e., monitoring of color at different humidity levels for Ac24 thin film on both types of substrates (Si wafer and glass covered with Au–Pd thin film) post-annealed at 60 °C. Figure 6 presents the calculated color coordinates (CIE X and CIE Y) of PVA films with 24% acetal content deposited on selected substrates for low (5% RH) as well as high humidity (95% RH). In the case of film on Si substrate, reflectance spectra are used for calculation, while for polymer film deposited on Au–Pd/glass, transmittance spectra at 5% and 95% RH are used. It is seen from Figure 6 that a substantial change of color takes place in the first case: the two points, for 5% and 95% RH, are well separated in the color space. On the contrary, when transparent substrate is used the change of the color due to humidity is not so distinct: the two points, associated with the sample colors at low and high humidity, respectively, almost overlap.

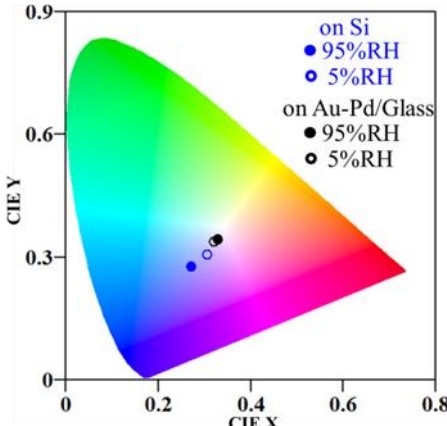

**Figure 6.** Calculated CIE coordinates for Ac24 thin film deposited on opaque (Si) and transparent (glass covered with Au–Pd thin film) substrates exposed to 5% and 95% RH.

## 4. Conclusions

The successful humidity-sensing application of thin films of hydrophobically modified PVA copolymers, namely poly(vinylalcohol-*co*-vinylacetal)s (PVA–Ac), of acetal content in the range of 18%–28% is demonstrated. A noticeable decrease of hysteresis, increase of sensitivity, and widening of dynamic range for modified films as compared to the neat PVA films are observed. For optimization of sensor performance, post-deposition annealing at 60 and 180 °C is used and two different film thicknesses are used (80 and 200 nm). The best sensor characteristics are obtained for films modified with acetal content around 24% with thickness of 80 nm and post-deposition annealing temperature of 60 °C. For relative humidity higher than 70% an accuracy of 0.7% RH is achieved. Both post-deposition annealing at 180 °C and higher film thickness (200 nm) leads to deterioration of sensing operation of the films.

It was demonstrated that both types of substrate used, silicon substrate and glass with thin (30 nm) metal (Au–Pd) overlayer, are suitable for humidity sensing. The first one is preferable if color sensing of humidity is considered, while the second one enables transmittance measurements thus offering more technological convenience and higher accuracy/resolution of measurements. For further decrease of hysteresis, a doping of PVA–Ac (24%) with $SiO_2$ particles (20%) is used. The thin film samples have full dynamic range and linear dependence of the measured signal in the entire humidity range. Humidity-sensitive films have thickness values around 80 nm that guarantees fast sensing.

**Author Contributions:** Conceptualization, T.B., D.C., and C.N.; methodology, T.B., D.C., and K.L.; software, T.B. and K.L.; validation, T.B., D.C., C.N., and K.L.; formal analysis, T.B., D.C., and K.L.; investigation, K.L. and S.B.; resources, D.C. and T.B.; data curation, K.L., D.C., and T.B.; writing—original draft preparation, T.B. and D.C.; writing—review and editing, T.B., D.C., and K.L.; visualization, T.B. and D.C.; supervision, T.B. and D.C.; project administration, T.B. and D.C.; All authors have read and agreed to the published version of the manuscript.

**Funding:** This research was funded by Bulgarian National Science Fund, Grant No. DN08-15/14.12.2016.

**Acknowledgments:** K.L. and S.B. acknowledge the National Scientific Program for young scientists and postdoctoral fellows, funded by Bulgarian Ministry of Education and Science (PMC № 271/2019). This work was partially supported by the European Regional Development Fund within the Operational Programme "Science and Education for Smart Growth 2014–2020" under the Project CoE "National center of mechatronics and clean technologies" BG05M2OP001-1.001-0008-C01.

**Conflicts of Interest:** The authors declare no conflict of interest. The funders had no role in the design of the study; in the collection, analyses, or interpretation of data; in the writing of the manuscript, or in the decision to publish the results.

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
