# Peer review of "Amphiphilic Poly(vinyl Alcohol) Copolymers Designed for Optical Sensor Applications—Synthesis and Properties"

_coatings, doi:10.3390/coatings10050460_

Round 1
Reviewer 1 Report
Good paper, well written. Couple of minor changes required.
Line 154, consider removing supplementary materials -do they add anything? If yes then include in the paper, if no then remove.
Line 161 - typo - post-deposition not posdeposition
Figures 1b,2b,3b,4c,4d, 5 & 6b . Unless this is integrated reflectance or transmittance then you should specify λmax in a separate table or because this is due to the frequency in modulation then even sample ID vs d (thickness) would help clarify the sample difference (won't just be ac%).
Conclusions
missing is a single line informing the reader of SiO2 nanoparticulate conclusions.
Otherwise - very nice.
Author Response
Reviewer #1
“Good paper, well written. Couple of minor changes required. Line 154, consider removing supplementary materials -do they add anything? If yes then include in the paper, if no then remove”.
Answer: We would like to thank to the reviewer for the positive evaluation of our work. We have included figure S1 from the supplementary materials in the revised manuscript (Figure 1(b) because we are confident that it adds useful information and illustrates the results in more comprehensive manner. Figure S2 has been removed because the data has already been included in the manuscript (Table 1)
“Line 161 - typo - post-deposition not posdeposition”
Answer: The misspelling has been corrected
“Figures 1b,2b,3b,4c,4d, 5 & 6b . Unless this is integrated reflectance or transmittance then you should specify λmax in a separate table or because this is due to the frequency in modulation then even sample ID vs d (thickness) would help clarify the sample difference (won't just be ac%)”.
Answer: Because this is not an integrated transmittance or reflectance we have included λmax for each sample in Table 2
“Conclusions
missing is a single line informing the reader of SiO2 nanoparticulate conclusions.
Otherwise - very nice”.
Answer: Conclusion has been revised in a way to be more expressive.
Reviewer 2 Report
The Authors report on composite polymer films for humidity sensing based on (optimized) mixtures of polyvinyl alcohol and polyvinyl acetal. Humidity is measured by optical methods based on the film deformation upon humidity absorption.
While the work describes experimental results in a rather broad range of humidity values, polymer mixtures and optimized thickness and compositions, my feeling is that the work lacks of a clear structure. First of all, experimental results are listed in a sort of chronological manner as one would expect in an activity report rather than in a scientific publication. I suggest authors to reformulate the work structure in a more easily exploitable format where specific questions are answered with the accurate design of experiment.
There is room to be more selective in choosing the most meaningful data and avoiding presenting results and then say that this material is not suitable for a given application.
Another issue that the work is very specific and it is not sufficiently put into a larger context, no comparison is made with other technologies or materials that would motivate this research or give emphasis to the results. Nanomaterials being a Journal with a broad readership, this element should be considered.
Graphics and tables are not particularly captivating. The work lacks of images of the device structure, SEM cross section images, DOE scheme.
Last thing, style editing by a native English speaker would facilitate the reading.
Author Response
“The Authors report on composite polymer films for humidity sensing based on (optimized) mixtures of polyvinyl alcohol and polyvinyl acetal. Humidity is measured by optical methods based on the film deformation upon humidity absorption.
While the work describes experimental results in a rather broad range of humidity values, polymer mixtures and optimized thickness and compositions, my feeling is that the work lacks of a clear structure. First of all, experimental results are listed in a sort of chronological manner as one would expect in an activity report rather than in a scientific publication. I suggest authors to reformulate the work structure in a more easily exploitable format where specific questions are answered with the accurate design of experiment.
There is room to be more selective in choosing the most meaningful data and avoiding presenting results and then say that this material is not suitable for a given application”.
Answer: We agree with the reviewer that the results were largely chronologically presented. Our idea was to show to the readers all steps of optimization. According to the reviewer’s comment, in the revised version of the manuscript, we have changed Figures 1, 2 and 3 and removed Figure 4. Additionally, we have rewritten entire section 3.2 selecting more precisely the meaningful data and avoiding less applicable results. Some changes have also been made in sections 3.3, 3.4 and 3.5.
“Another issue that the work is very specific and it is not sufficiently put into a larger context, no comparison is made with other technologies or materials that would motivate this research or give emphasis to the results. Nanomaterials being a Journal with a broad readership, this element should be considered.”
Answer: We disagree with the reviewer that the work is very specific. The main output of the study is the optimized humidity sensitive material that can be applied in diverse sensor types. In the manuscript we have demonstrated as examples two possible ways of transducing the films dimensional changes: measuring reflectance / color of single sensitive film deposited on planar absorbing substrate or measuring transmittance of film deposited on planar glass substrate with metal overlayer. However, the application of our sensitive copolymer films can be further extended. They can be utilized as sensitive coatings in optical fibers, for example.
We disagree with the reviewer that the manuscript lacks of comparison with other materials. The aim of the research was to enhance further the good humidity sensing properties of neat PVA. Two approaches were suggested: doping with SiO2 particles and partial acetalization. The properties of modified PVA films were compared with neat PVA and enhancement was clearly demonstrated. According to the justification of PVA as humidity sensitive materials, the entire introduction is devoted to this task.
“Graphics and tables are not particularly captivating. The work lacks of images of the device structure, SEM cross section images, DOE scheme”.
Answer:
As we have already mentioned Figures 1, 2 and 3 have been revised. SEM images of the cross-section of the samples have not been made because there is no a profile or structure. We do not think that SEM cross-section images will add valuable information because we use planar substrates that have no surface relief or diffraction pattern.
“Last thing, style editing by a native English speaker would facilitate the reading”.
Answer: The manuscript has been edited by a fluent English speaker
Round 2
Reviewer 2 Report
I can see a real effort from the Authors to aswer my comments, my opinion is that the work is now suitable for publication.
Please check the Figure 1 image resolution.
Regards